# Does joint impedance improve dynamic leg simulations with explicit and implicit solvers?

**Serhii Bahdasariants**[1], **Ana Maria Forti Barela**[2], **Valeriya Gritsenko**[1,3], **Odair Bacca**[2], **José Angelo Barela**[4], **Sergiy Yakovenko**[1,3,5,6]*

1 Department of Human Performance, School of Medicine, West Virginia University, Morgantown, WV, United States of America, 2 Institute of Physical Activity and Sport Sciences, Cruzeiro do Sul University, São Paulo, SP, Brazil, 3 Department of Neuroscience, School of Medicine, West Virginia University, Morgantown, WV, United States of America, 4 Department of Physical Education, São Paulo State University, Rio Claro, SP, Brazil, 5 Department of Mechanical and Aerospace Engineering, Benjamin M. Statler College of Engineering and Mineral Resources, West Virginia University, Morgantown, WV, United States of America, 6 Department of Chemical and Biomedical Engineering, B.M. Statler College of Engineering and Mineral Resources, West Virginia University, Morgantown, WV, United States of America

* seyakovenko@mix.wvu.edu

**Data Availability Statement:** Data and scripts will be published in the public repository on GitHub.

**Funding:** SY #1 R03 HD099426-01A1, NIH; AB, VG, SY #2018/04964-8, FAPESP. The sponsors played no role in the study design, data collection

## Abstract

The nervous system predicts and executes complex motion of body segments actuated by the coordinated action of muscles. When a stroke or other traumatic injury disrupts neural processing, the impeded behavior has not only kinematic but also kinetic attributes that require interpretation. Biomechanical models could allow medical specialists to observe these dynamic variables and instantaneously diagnose mobility issues that may otherwise remain unnoticed. However, the real-time and subject-specific dynamic computations necessitate the optimization these simulations. In this study, we explored the effects of intrinsic viscoelasticity, choice of numerical integration method, and decrease in sampling frequency on the accuracy and stability of the simulation. The bipedal model with 17 rotational degrees of freedom (DOF)—describing hip, knee, ankle, and standing foot contact—was instrumented with viscoelastic elements with a resting length in the middle of the DOF range of motion. The accumulation of numerical errors was evaluated in dynamic simulations using swing-phase experimental kinematics. The relationship between viscoelasticity, sampling rates, and the integrator type was evaluated. The optimal selection of these three factors resulted in an accurate reconstruction of joint kinematics (err < 1%) and kinetics (err < 5%) with increased simulation time steps. Notably, joint viscoelasticity reduced the integration errors of *explicit methods* and had minimal to no additional benefit for *implicit methods*. Gained insights have the potential to improve diagnostic tools and accurize real-time feedback simulations used in the functional recovery of neuromuscular diseases and intuitive control of modern prosthetic solutions.

## Introduction

The central nervous system (CNS) evolved to orchestrate muscular and skeletal actions to produce complex motor behaviors [1]. For instance, in locomotion, intrinsic and sensory feedback

and analysis, decision to publish, or preparation of the manuscript.

**Competing interests:** The authors have declared that no competing interests exist.

signals are integrated with descending visuomotor commands to fine-tune limb stepping and foot placement [2, 3]. To explain this intricate interplay between neural and mechanical systems, previous research has produced a variety of biomechanical models and analyzed their interactions with the environment [4]. For example, a biomechanical model of the arm was shown to explain the directional tuning property of the primary motor cortex neurons, describing why some cortical neurons are most active in particular movement directions [5]. Similar models of neuromechanical interactions are believed to exist within neural pathways in order to overcome transmission delays and nonlinear limb dynamics [6]. The use of embedded dynamical computations for motor control is illustrated in Fig 1. The *desired* motion transformed through the inverse dynamics yields control signals to muscles (Fig 1A). The oversight over the execution of these motor commands can also be monitored with the use of embedded forward simulations (Fig 1B). Because biomechanical computations can be expressed within neural pathways, models of musculoskeletal transformations are *biomimetic* and may capture mechanistic cause-effect relationships.

Gait abnormalities can be diagnosed and treated more effectively using forward dynamic simulations of body motion [7–9]. However, subject-specific pathomechanics may be required for effective rehabilitation [10, 11]. Simple models may capture the dynamics of the limb in real time, but they may fail to capture the muscle-specific actions that require musculoskeletal morphology [12–18]. Conversely, detailed models with a minimum of 6 degrees of freedom (DOF) and 43 major muscles may require more time to execute [19]. Thus, the engineering challenge in biomechanical modeling is to balance model complexity and computational efficiency. For this reason, we have previously modified Sartori et al. [20] approximation method to derive an objective approximation of muscle dynamics to accurately describe arm and hand muscles in the context of real-time simulations with reduced computational load [21, 22]. This development can be potentially extended to model the muscle dynamics of the leg [23]. However, the fast and accurate simulations of *body dynamics* remain a challenge [24, 25].

Recently, the use of efficient physical engines [26] has been combined with the musculoskeletal models [19] to develop a new iteration of tools for research, health, and consumer applications [27]. Physical engines use various numerical methods (e.g., Runge-Kutta, implicit/semi-implicit Euler) to integrate ordinary differential equations (ODE) describing multi-DOF body dynamics. Accurate solutions are often achieved by selecting the integration

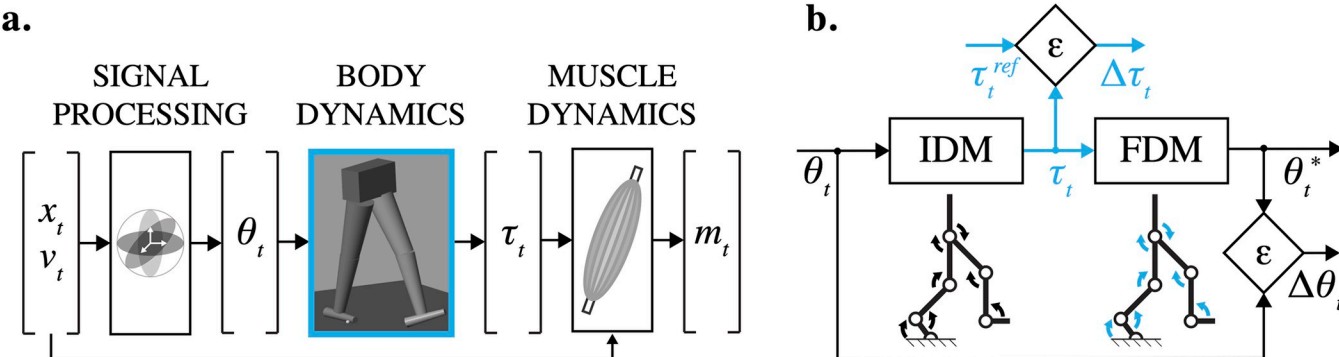

**Fig 1.** Inverse model of the human motor control (a) and the model validation pipeline (b). From left to right: (1) signal processing—the joint angles $\theta_t$ are found from kinematic equations given the limb positions $x_t$; (2) body dynamics—the joint torques $\tau_t$ are computed using motion equations, given joint angles and their derivatives; (3) muscle dynamics—the muscle forces and computed muscle activations $m_t$ are calculated using joint torques ($\tau_t$), limb positions ($x_t$), velocities ($v_t$), and musculoskeletal geometry. The model validation process: (1) torques $\tau_t$, computed using inverse dynamics model (IDM), are compared to reference torques $\tau_t^{ref}$ to validate the solution (error: $\Delta\tau_t$); (2) joint angles $\theta_t^{*}$, simulated with forward dynamics model (FDM), are compared to the experimental angles $\theta_t$ to confirm the cause-and-effect relationship between the $\tau_t$ and $\theta_t$ (error: $\Delta\theta_t$).

time step ($\Delta t$) that satisfies a set tolerance error. Namely, the time step is decreased for rapid system dynamics to maintain integration accuracy and increased for the slow dynamics to speed up the simulation. In real-time decoding, for example, in robot-assisted applications [28], the computing power provides just-in-time or even faster than real-time accurate calculations for movement planning. What is especially challenging is the inclusion of object manipulation in these simulations. Erez et al. compared the performance of real-time engines to demonstrate that each engine is currently optimized for its corresponding use (gaming or robotics) through the difference in system formulation (Cartesian vs. generalized coordinates) and body internal/external forces calculation methods [29]. However, not surprisingly, all simulations fail with the **increase** in the simulation time step, which indicates a challenge for applications with low sampling rates. Yet, the growing virtual reality use in the post-stroke rehabilitation [30, 31] further necessitates computing body dynamics accurately and swiftly [32]. The use of motion capture systems to acquire body kinematics in these interactive environments is typically performed with fixed sampling frequencies, linked to the low display refresh rate (60 or 120 Hz). Thus, there is a need to improve simulation stability at low sampling rates where numerical inaccuracies result in transient kinematic perturbations.

In this study, we developed a biomechanical framework for testing viscoelastic joint constraints and numerical solver types to reduce simulation errors at low sampling rates. We posited that the transient kinematic perturbations due to the large integration time step could be smoothed by adding viscoelastic joint impedance to increase simulation stability. Furthermore, we hypothesized that the viscoelastic compensation depends on the type of a used numerical solver. We tested these hypotheses in a human bipedal model simulating locomotion. We used three numerical solvers with varied combinations of joint impedance to simulate the inverse and forward dynamics of the swing locomotor phase. We determined optimal viscous and elastic pair that stabilized the kinematic solutions and produced close-to-physiological kinetics. In addition, we evaluated feasibility of our approach for real-time applications.

## Materials & methods

This study aimed to test the stabilization effects of viscoelastic compensation, evaluate the performance of numerical solvers, and identify sampling frequencies that produce accurate and stable simulations of the leg dynamics. This was accomplished with the following four steps to develop: (1) subject-specific morphometry scaling of the inertial properties of the body segments, (2) forward and inverse computations for simulating kinematics and kinetics, (3) search for optimal impedance and simulation rate, and (4) validation pipeline for computing numerical errors.

### Kinematic and kinetic datasets

Three datasets were used for the development and validation of the inverse and forward dynamics: (1) experimental marker trajectories, (2) joint angles, and (3) joint torques simulated during the swing locomotor phase.

### Dataset 1: Marker trajectories

The representative three-dimensional marker trajectories were recorded for one healthy individual (male, 26 y/o, 72 kg, 1.75 m). The Institutional Review Board of Cruzeiro do Sul University approved all the procedures, and the participant signed a written consent form prior to the experimental session (institutional review board (CAAE) 02887518.2.0000.8084). A computerized gait analysis system (Vicon, Oxford Metrics, Inc.) with eight infrared cameras tracked reflective markers placed bilaterally on the anterior superior iliac spine, posterior superior iliac

spine, medial and lateral epicondyles of the femur, tibialis tuberosity, medial and lateral malleoli, inter-malleolus, and first and fifth metatarsals, following the recommendations of the International Society of Biomechanics [33]. Prior to data acquisition, a T-pose kinematic calibration trial was performed with all reflective markers to register marker clusters relative to the anatomical landmarks. Subsequently, markers were removed from the medial and lateral epicondyles of the femur, tibialis tuberosity, medial and lateral malleoli, and inter-malleolus. The subject was asked to walk at a self-selected comfortable speed without interruption along a 10-m walkway equipped with two force platforms (Kistler, Model 9286BA). Practice was allowed before the data recording session to familiarize the subject with the laboratory environment and the walking condition. From the five recorded trials, 10 unilateral swing phases were extracted using heel-strike and toe-off events, identified based on experimental ground reaction forces and vertical velocity of the foot [34].

### Dataset 2: Joint angles

The hip, knee, and ankle joint angles, as well as the standing foot rotation with respect to the ground, were calculated from the *Dataset 1* using the biomechanics analysis package Visual3D (C-Motion, Inc.). The data were low-pass filtered using a 2$^{nd}$ order Butterworth filter with the cutoff frequency of 6 Hz and resampled with cubic splines [35] at $f$ = 50, 100, 200, 300, 400, 500, 2000 Hz.

### Dataset 3: Joint torques

Joint torques were simulated from the computed joint angles (*Dataset 2*). Since the numerical solution approaches analytical as the sampling rate increases, torques calculated from angles sampled at 2 kHz were used as a reference ($\tau_t^{ref}$) for all torques simulated at lower frequencies with and without viscoelastic impedance. This impedance was modeled as varied combinations of rotational stiffness and damping coefficients.

### Model and anatomical scaling

A bipedal biomechanical model describing the movement of 17 DOFs was assembled to solve inverse and forward dynamics in Simscape Multibody (R2022b, MathWorks). The proximal to distal segments of the human leg were represented by solid frustums of right circular cones (wide base placed proximally) using subject-specific inertial properties based on validated mathematical relationships [36]. This refinement reduced inertial representation errors to <10% of observed properties. The centers of mass, locations of the hinge points, and inertia were derived for each body segment using Hanavan's formulations (see S1 Fig). The segment weight was calculated as a fraction of the total human body weight using Winter's tables [37, p. 86]. The weight of upper body was assigned to the pelvis segment. The axial length of the lower body parts was expressed as a fraction of the subject's height. The segment geometry was adjusted according to sex and age. Thigh and shank proximal base diameter (*d*) were calculated from a circumference equation $C = \pi d$, where the circumference $C$ was fetched from the sex- and age-based anthropometric reference tables [38, 39]. The proximal base diameter for a foot segment was estimated as a malleolus height. The distal base diameter of all segments was found from the regression equations describing the relationship between axial length and base diameters [36]. The motion of rotational DOFs between the base and follower bodies was expressed as a sequence of time-varying rotational transformations. The knee joint was modeled with a simple 1 DOF revolute primitive. The ankle and hip joints were modeled as a combination of a revolute and universal 2 DOF primitives to avoid the gimbal lock problem (undesired alignment of two perpendicular axes). The foot was rigidly connected to the ground

(no slip condition for the leg in stance) close to the metatarsophalangeal (MTP) joint (75% of foot length), and MTP was allowed to rotate in 3 DOFs. This modeled an approximate location of the foot center of pressure during the single stance phase [40]. Since the analysis was focused on the leg in swing, the motion of leg in stance was determined by its kinematics in both forward and inverse simulations. This hybrid design prevented kinetic errors from propagating from the standing to the swinging leg. While 17 DOFs were simulated, we focused our analysis on the seven DOFs of the leg in swing.

## Body dynamics simulations

A built-in recursive algorithm (Simulink R2022b) was used to simulate the inverse dynamics of the modeled body; the explicit 4th-order Runge-Kutta and explicit/implicit Euler methods were used to simulate the forward dynamics. Euler *explicit* integration is the least accurate and yet the fastest. Euler *implicit* formulation is better suited to keep the integration error bound to the tolerated error; however, it is slower than the *explicit* type [41, p. 58]. The accuracy and speed of the Runge-Kutta method are somewhere between those of the two previous methods. For all fixed-step solvers, the integration step is inversely proportional to the sampling rate *f*. Fig 1B describes the simulation-validation pipeline, where the joint angles ($\theta_t$, *Dataset 2*) are calculated from the experimental marker trajectories (*Dataset 1*) and then used in the inverse dynamics model (IDM) to simulate joint torques ($\tau_t$, *Dataset 3*). These torques propagate through the forward dynamics model (FDM) to simulate joint angles ($\theta_t^*$). Simulated and observed angles were compared to assess the accuracy. The kinetic ($\Delta\tau_t$) and kinematic ($\Delta\theta_t$) errors were calculated as root-mean-square errors (RMSE) between simulated ($\tau_t$, $\theta_t^*$) and reference / observed signals ($\tau_t^{ref}$, $\theta_t^{ref}$). A dynamic translation technique (*Accelerator Mode*, Simulink R2022b) allowed us to change input signals without terminating the model execution and increased the simulation efficiency. A standard portable computer running macOS was used for all simulations.

## Impedance optimization

Passive joint impedance was described with the rotational stiffness and damping *(k,b)* of the spring-dampers used to stabilize multibody simulations [42]. As expected, high impedance not only decreased angular errors but also increased torque errors. This, therefore, presented a sensitivity problem, which involved identifying optimal impedance values to reduce simulation errors. To find the optimal impedance, we searched for the minimal integrated kinematic and kinetic error. Computed kinetic ($\Delta\tau_t$) and kinematic ($\Delta\theta_t$) errors for each DOF were normalized to peak-to-peak torque and angle ranges, respectively. We covered the grid of multiple $k \in [0, 10^{-4}, 10^{-3}, 10^{-2}, 10^{-1}]$ Nm/deg and $b \in [0, 10^{-4}, 10^{-3}, 10^{-2}, 10^{-1}]$ Nm/deg/s for multiple experimental swing phases, simulation rates, and numerical solvers:

*Normalization for each DOF, solver type, and sampling frequency*:

$$A = \Delta\theta^{[N \times K \times B]} / \theta_{pp}^{[N \times 1]}; \ T = \Delta\tau^{[N \times K \times B]} / \tau_{pp}^{[N \times 1]}, \tag{1}$$

Optimization:

$$(k, b)^{[D \times S \times F]} = min(A^{[K \times B \times D \times S \times F]} + T^{[K \times B \times D \times S \times F]}), \tag{2}$$

The superscripts denote dimensionality. *N* is the number of swing phases; *K* is the number of stiffness coefficients; *B* is the number of damping coefficients; *D* is the number of DOFs; *S* is the number of solvers; *F* is the number of simulation rates; $\theta^{pp}$ and $\tau^{pp}$ are vectors containing joint range of motion and torque peak-to-peak values, respectively; and *(k,b)* is optimal stiffness and damping matrix.

## Statistical analysis

The normality was tested with the Anderson-Darling test for the following variables: model execution time, kinematic and kinetic errors [43]. The appropriate statistical significance level (α) initially set to 5% was adjusted with the Holm-Bonferroni correction, and the difference between simulations with and without the viscoelastic contribution was tested with the Wilcoxon rank sum test. The linear regression demonstrated the correlations between the execution times and simulation rates. All analyses were performed using the Statistics and Machine Learning Toolbox (MATLAB, R2022b, MathWorks Inc.)

## Results

In this study, we used viscoelastic joint impedance to reduce numerical inaccuracies in simulations of limb dynamics. The impedance, increasing from proximal to distal joints, significantly decreased errors of explicit integration. The beneficial impedance stabilization was negligible when used with the implicit Euler method, which outperformed the explicit Euler and Runge-Kutta methods. The added impedance generated an expected increase in torque errors, however, stabilized kinematics at low sampling rates. Overall, the simulation errors were below 1% of peak-to-peak kinematics, and below 5% of peak-to-peak kinetics. All simulations were executed faster than in real time.

### Optimal viscoelastic impedance

We modified viscoelastic joint impedance to determine the optimal values that minimize both kinetic and kinematic errors in simulations of leg movements. High impedance values increase torque errors (Fig 2A) and decrease kinematic errors (Fig 2B); low impedances have the opposite effect (the results for all DOFs and solver types are demonstrated, S2 and S3 Figs). We, therefore, chose the optimal stiffness and damping sets as generating minimal kinetic and kinematic errors at a given sampling rate. These sets were different for different DOFs and increased in value from proximal to distal joints along the limb, i.e., the viscoelastic contribution was the lowest at the hip and the highest at the ankle. These values are shown with the blue outline in Fig 2 (see S1 and S2 Tables in S1 File contain this information for all DOFs in the model).

The improvements in kinematic accuracy with additional impedance were not uniform across three types of numerical solvers. For example, Fig 3A shows no significant difference for kinematic errors in ankle flexion-extension computed with and without impedance across three solvers. However, adding impedance to *explicit* integration reduced the tendency to numerically "overshoot" exact kinematic solutions in 6 out of 7 analyzed DOFs in the swinging leg. Fig 3B and 3C demonstrate this beneficial improvement in the solution accuracy for the knee flexion-extension and hip abduction-adduction DOFs. The viscoelastic contribution to the *implicit* Euler method was unnecessary for 5 out of 7 modeled DOFs, with one of these DOFs (internal-external rotation of the hip) benefiting from stabilization only at low sampling frequencies (50–100 Hz). And yet, reinforcing the implicit Euler method with viscoelastic stabilization reduced errors in ankle internal-external rotation and hip adduction-abduction kinematics. Fig 6 shows that the kinetic profiles of these DOFs have frequent sign-switching slopes. These led to higher integration errors and the need for stabilization. Adding impedance to these joints reduced kinematic errors (Fig 3C) but increased kinetic errors. Fig 4A shows that with added impedance, the torque errors increase moderately (marked with *). While the kinetic errors with and without impedance were significantly different in 4 out of 7 examined DOFs, they did not differ at low sampling frequencies in some DOFs. These included ankle flexion-extension (50–100 Hz), its eversion-inversion (50 Hz), and knee flexion extension (50–

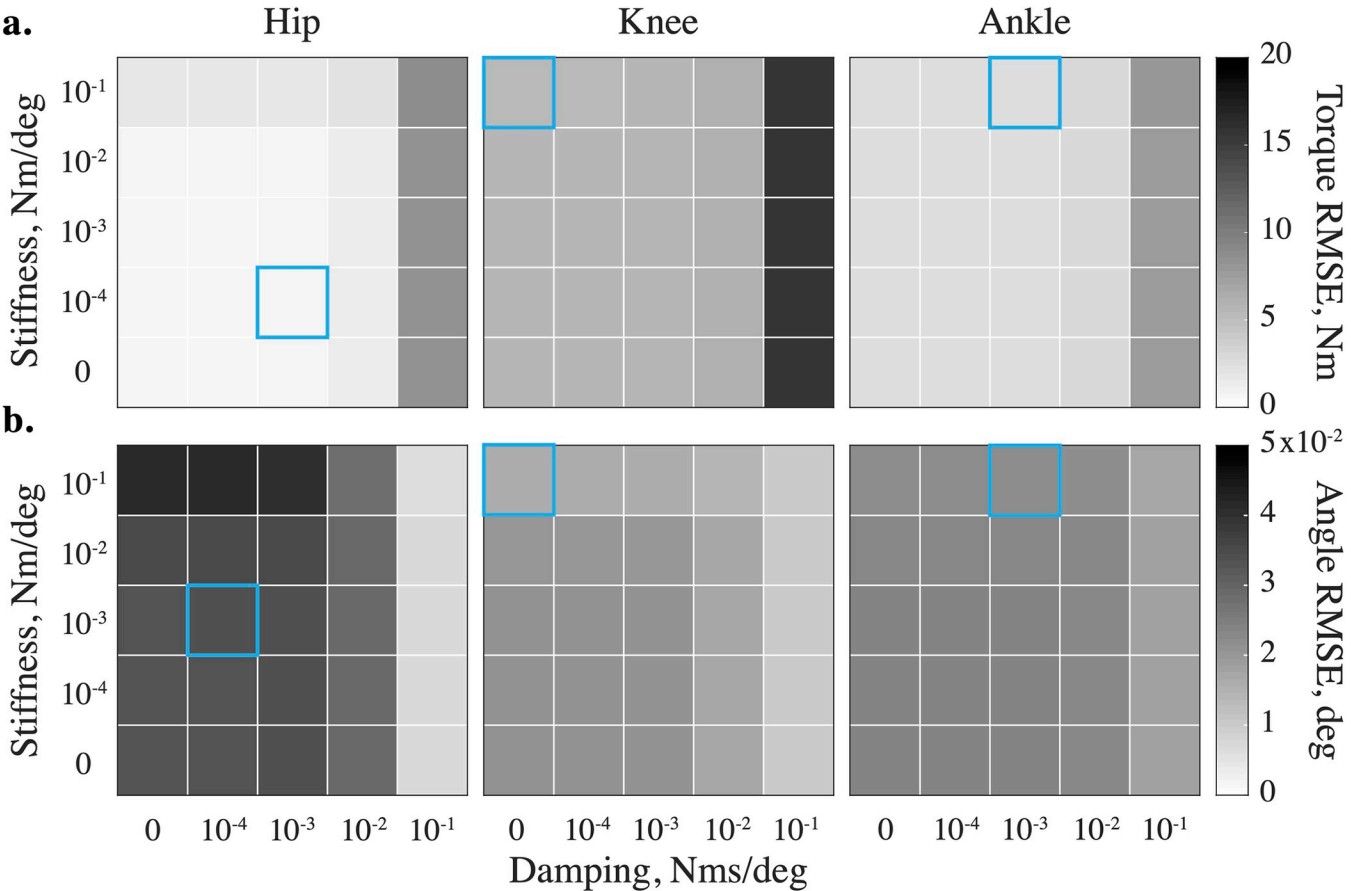

**Fig 2. Stabilizing impedance affects (a) kinetic and (b) kinematic errors differently across DOFs.** The errors are shown for the flexion-extension DOFs of the swinging leg's hip, knee, and ankle during one representative swing phase simulated with the build-in recursive (a) and implicit Euler (b) methods at 200 Hz. The kinetic performance decreases at high values, and the kinematic performance decreases at low values. The errors obtained with optimal ($k$,$b$), specific to DOF, sampling frequency, and solver, are squared in light blue.

200 Hz) for which relatively high stiffness (up to $10^{-1}$ Nm/deg) or/and damping (up to $10^{-1}$ Nm/deg/s) were applied to counter integration inaccuracies in forward simulations (see S1 and S2 Tables in S1 File for details). The viscoelastic contribution was therefore effective for forward simulations with explicit methods and negligible for inverse simulations with fixed low sampling rates. In general, the use of stabilizing impedance was disadvantageous in inverse simulations with high sampling rates.

### Numerical solvers

The choice of implicit or explicit numerical integration largely determined the accuracy of the forward simulations. The implicit Euler method outperforms explicit methods as demonstrated by low kinematic errors in Figs 3 and 4B. S4 Fig provides further details of the comparative performance for all joints. The best simulation accuracy even at low sampling rates was achieved by the implicit Euler method.

### Model execution time

The forward and inverse dynamic models were executed faster than in real time across all tested sampling frequencies. Model execution times were normalized to the respective swing

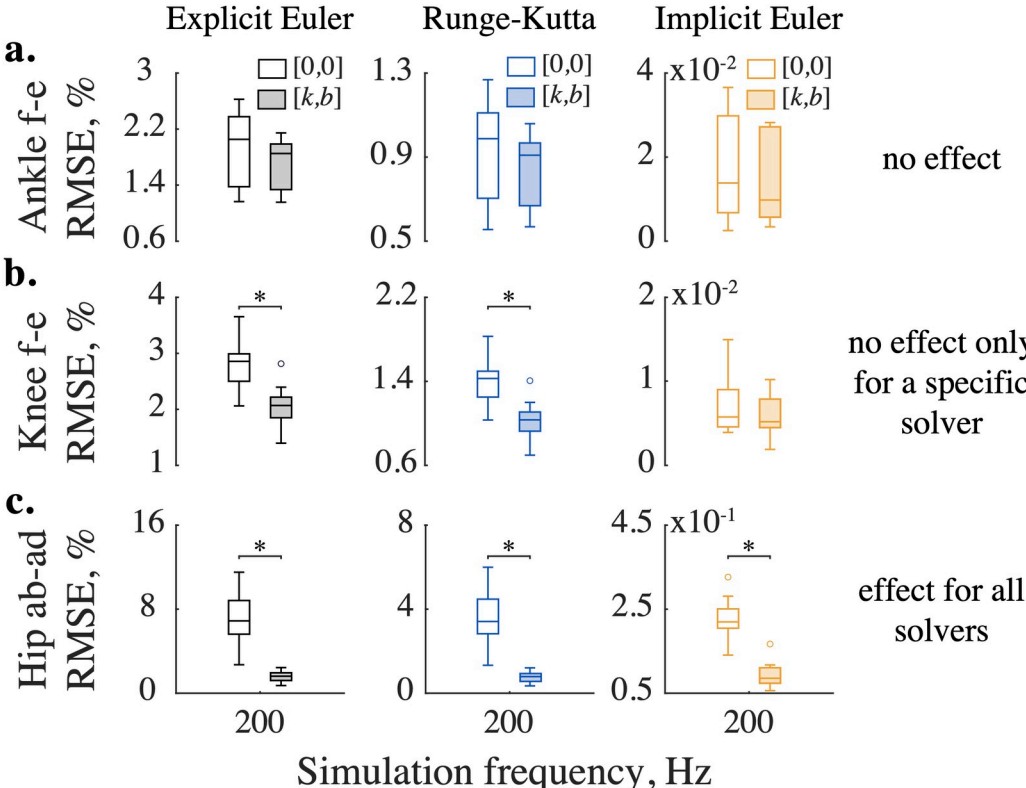

**Fig 3. Stabilizing impedance affects kinematic errors differently across solvers.** Demonstrated are three representative cases when impedance had: (a) no stabilizing effect across solvers, (b) no effect for a specific solver, and (c) effect for all solvers. The error distributions are shown for ten swing phases simulated at the sampling frequency of 200 Hz. The bottom and top edges of the box plots show the 25th and 75th percentiles of the error distributions, horizontal lines demonstrate medians, whiskers extend to maximum errors, and outliers are marked with circles. Empty boxes represent distributions computed without impedance, while shaded boxes correspond to errors obtained with optimal impedance $(k,b)$ specific to DOFs and solvers. The horizontal bars indicate significant differences as tested with the Wilcoxon signed-rank test and corrected with the Holm-Bonferroni method $(p < 1.3 \times 10^{-3})$.

phase duration. Fig 5 shows the relationship between simulation frequency and execution speed. The real-time speed metric was expressed as the ratio of median values of simulation time (N = 10) to swing durations, i.e., values below 1 are faster than real time. The regression indicates the expected linear increase in execution time with the increase in the sampling rate. The gray dashed lines indicate performance multiples relative to real time. For example, the execution of the inverse model is about 150x and 35x faster than real-time at 50 Hz and 500 Hz, respectively. Due to the difference in algorithmic complexity of integration and differentiation methods, forward simulations are slower than inverse ones across all sampling frequencies. Yet, the forward simulations of the leg model were faster than real time. At 500 Hz, the explicit Euler method (~9x) was faster than the Runge-Kutta (~7x) and implicit Euler (~3.5x) methods.

## Kinetic and kinematic trajectories

Lower limb dynamics was simulated with forward and inverse models. Fig 6 shows joint torques computed with the built-in recursive solver (A) and respective joint angles calculated with the implicit Euler method at 200 Hz (B). The variability of swing dynamics is depicted with standard deviation from the mean (see shaded areas for ±s.d. in Fig 6). The computed

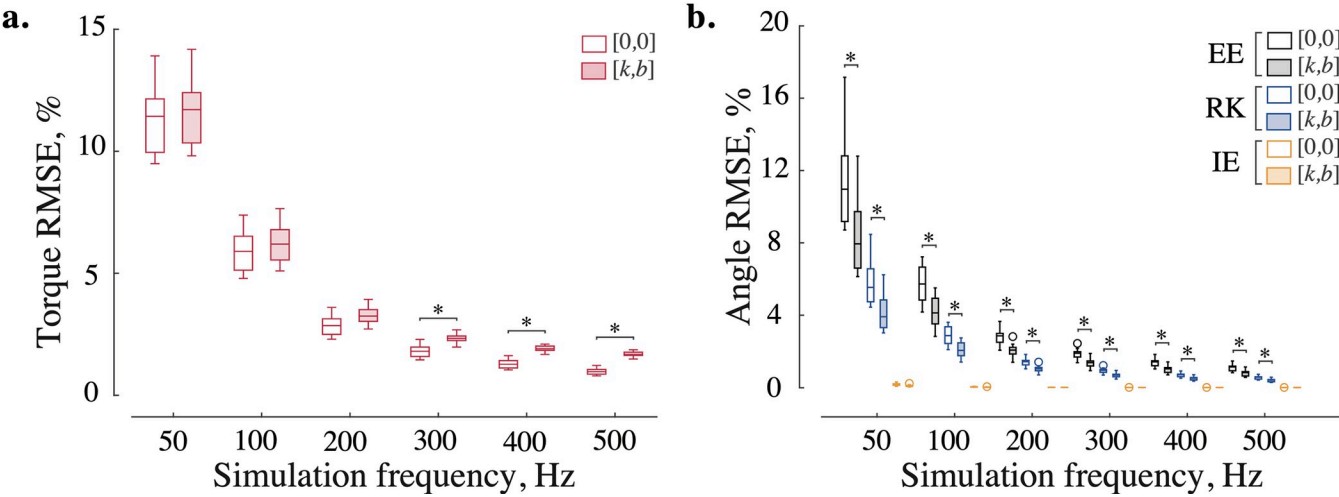

**Fig 4.** Viscoelastic impedance affects inverse (a) and forward (b) dynamic errors across frequencies. The built-in recursive algorithm and three numerical integrators become more accurate with an increasing sampling frequency (empty boxes); viscoelastic impedance increases kinetic and decreases kinematic errors (shaded boxes), as shown for ten swing phases of the knee flexion-extension DOF. For the sampling rates over 200 Hz, kinetic error distributions with and without impedance were significantly different ($p < 1.83 \times 10^{-4}$); the kinematic distributions computed with explicit methods were different across all sampling rates, while the errors obtained with implicit method were the same ($p < 1.3 \times 10^{-3}$). All distributions were compared using the Wilcoxon signed-rank test and corrected with the Holm-Bonferroni procedure. The description of the boxplots is the same as in Fig 3. Abbreviations in legend: EE—explicit Euler, RK—4th order Runge-Kutta, and IE—implicit Euler methods.

variability is similar to other studies [44–46]. The form and absolute values of the computed average signals are also similar to those demonstrated in previous studies [47–49, 50, p. 165]. We selected a 200 Hz simulation frequency to keep kinematic and kinetic errors within 1% and 5% tolerance, respectively (S4 Fig). Generally, implicit methods do not improve accuracy with the additional impedance. Only explicit methods benefit from it.

## Discussion

In this study, we developed a real-time biomechanical framework for testing viscoelastic parameters that reduce the dynamic simulation errors in a human dynamic model of locomotion. The development involved (1) subject-specific body inertia scaling, (2) forward and inverse dynamic computations for simulating kinematics and kinetics, (3) a search for optimal impedance and simulation frequency, and (4) numerical validation and error assessment. Height, weight, age, and sex parameters were used for scaling segment inertia. The simulation errors were small in forward ($< 1\%$) and inverse ($< 5\%$) simulations as computed with implicit Euler and built-in recursive methods, respectively (see also S4 Fig). The inaccuracies in kinematics associated with explicit solvers matched values found in prior research on human locomotion [51, 52]. Added viscoelastic impedance increased the precision of explicit simulations but, in general, decreased the physiological plausibility of the forces computed in inverse simulations. Nevertheless, small errors could be tolerated as they do not surpass the inaccuracies observed in previous studies [7, 53] and result in increased integration accuracy with explicit methods, particularly when the frequency of simulations is low (Fig 5). The implicit integration method outperformed all explicit methods and required no additional compensation from the impedance. Since explicit methods are standard in physics engines, the use of joint impedance could improve their numerical stability; however, implicit methods deliver the best results. This is supported by the comparative demonstration of handling object interactions with standard physics engines and the MuJoCo engine, which utilizes the implicit integration of velocity [26].

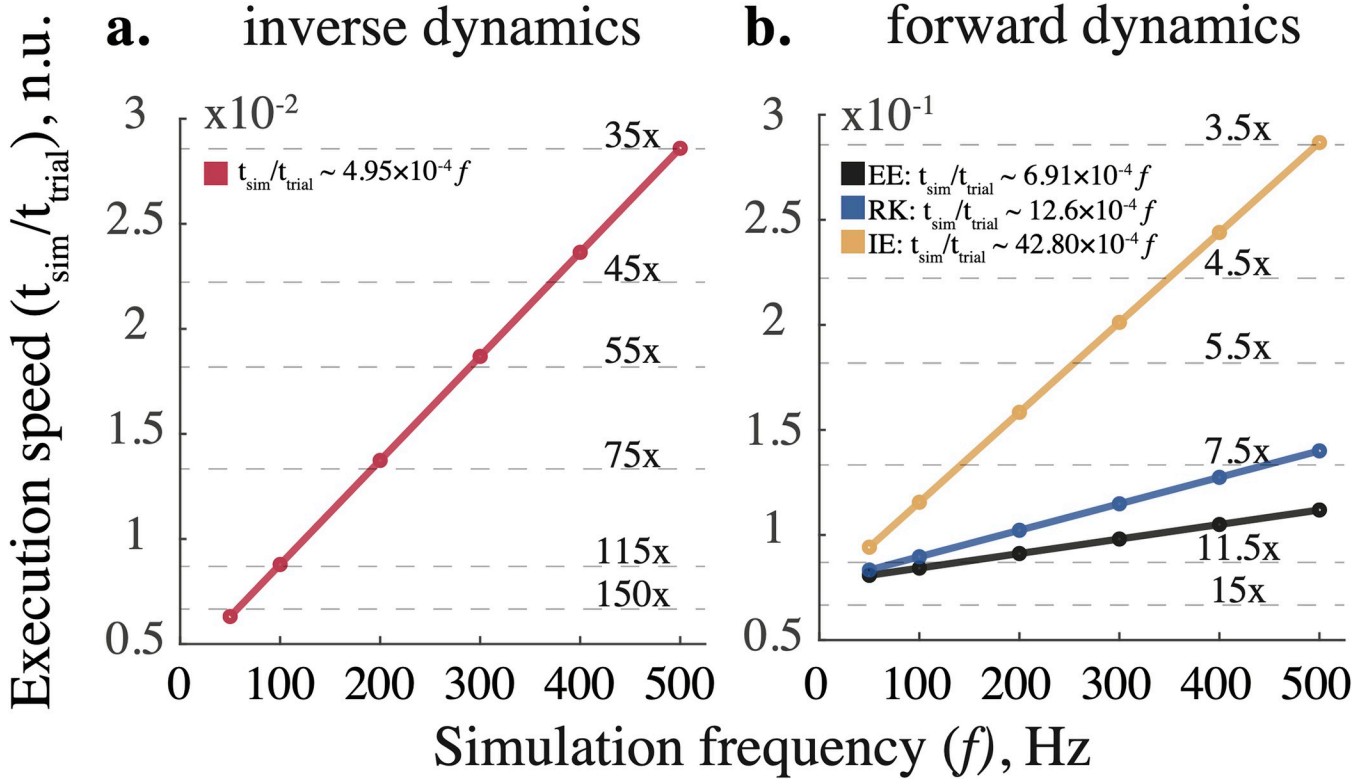

**Fig 5.** Model execution speed scales linearly with simulation frequency for (a) inverse and (b) forward dynamic simulations. Built-in recursive solver (shown in red), explicit Euler (EE), Runge-Kutta (RK), and implicit Euler (IE) methods demonstrate faster than real-time performance across tested simulation frequencies. Circles (o) represent the median execution times registered and normalized to the swing phase duration for each simulation. The thin dashed lines indicate performance multiples relative to the real time (when simulation and experimental time are equal). Thick lines illustrate linear regressions; all R-squared values are greater than 0.9887. The equations describing an approximate relationship between the two variables are expressed in legends. Abbreviations in legend are the same as in Fig 4.

### Integration method

Accurate dynamic simulations of human limbs in real time require an efficient integration method. Analytical solutions provide the best computational efficiency; however, they may not exist for complex mechanical systems requiring fixed- or varied-step numerical methods to simulate body movement [54, p. 5]. Fixed-step methods are typically used to avoid a sample mismatch between input data acquisition and simulation rates; however, they suffer from the accumulating truncation error. Variable-step methods are challenging to implement in real time because they require a predictive extrapolation of real-time inputs and outputs in simulation. The integration methods of approximating ODE can be explicit or implicit. Explicit schemes compute the future state of the system ($t+\Delta t$) from its current state ($t$). However, they are often unstable when simulating 'stiff' dynamical systems with slow- and fast-changing dynamical variables [55, p. 614]. For example, the knee swiftly extends during the mid-swing and flexes slowly at the end of the swing of a human gait cycle; the time of ground contact also introduces a sharp transition between swing and stance. Implicit ("backward") schemes compute approximations using both the current and future states [41, p. 55]. This requires solving for unknowns using a root-finding algorithm (for example, Newton's method). The additional computational complexity within implicit methods is relatively less taxing than the advantage of accuracy, as shown in Fig 5.

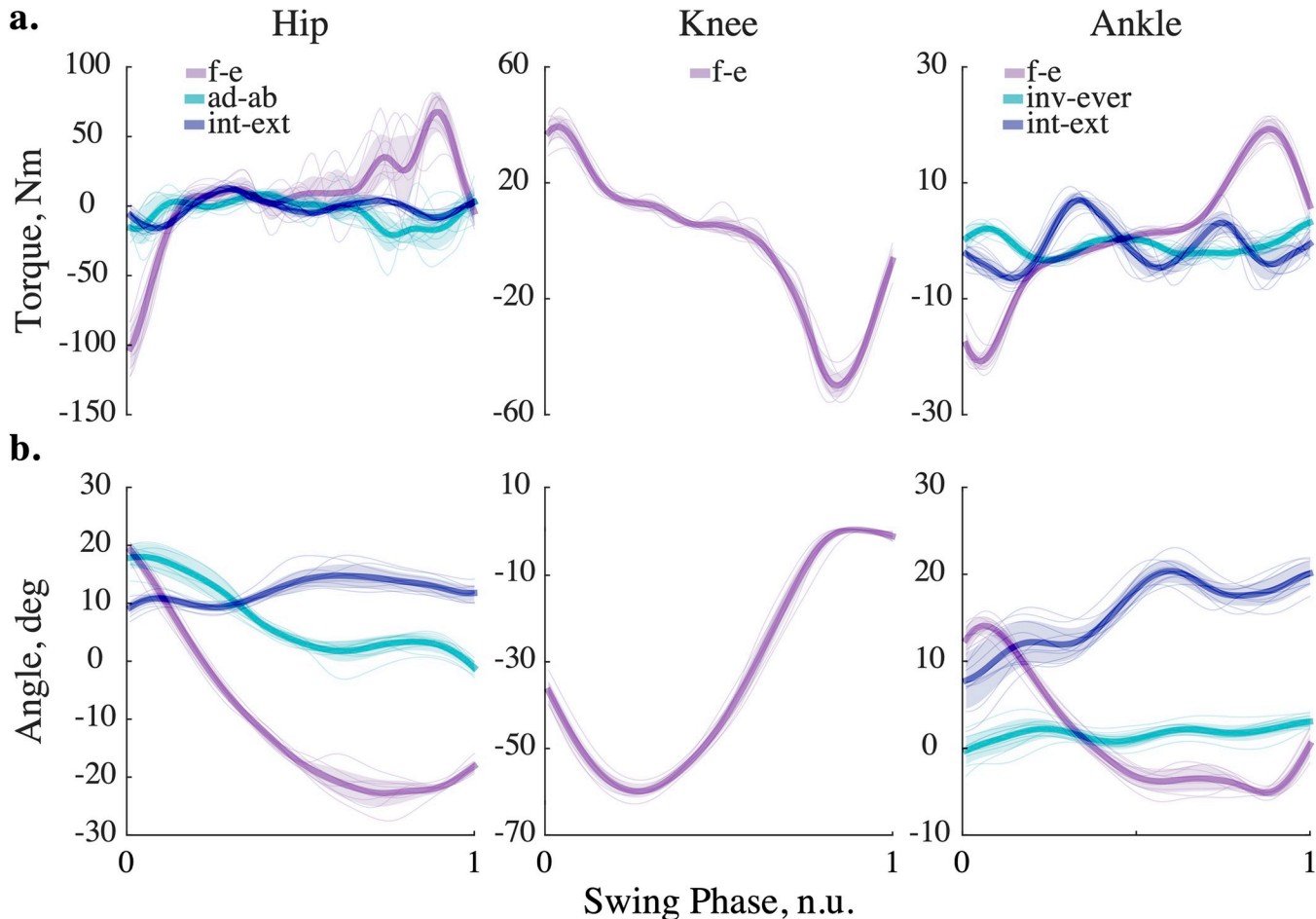

**Fig 6. Simulated dynamics of the leg in the swing.** (a) Kinetic trajectories, simulated at 200 Hz with implicit Euler method and no viscoelastic contribution are shown for ten swing phases. The thick lines correspond to the averages; the shaded areas show ± standard deviation; the thin lines correspond to single trials. (b) Kinematic trajectories; the description is the same as in a. Abbreviations: f-e—flexion-extension; ad-ab—adduction-abduction; int-ext—internal-external rotation; inv-ever—inversion-eversion; in each legend entry, the first movement type corresponds to the negative direction on the vertical axis and the second movement type corresponds to the positive direction.

We selected the simplest implicit integrator—the implicit Euler method—to solve the forward dynamics of the human leg in swing. Other integrators often used for physical simulations include explicit/semi-implicit Euler and (explicit) 4-th order Runge-Kutta (RK) [56, 57]. The semi-implicit Euler employs a hybrid integration scheme to solve dynamical systems and is an improvement over the least accurate, explicit Euler method. Used in the context of biomechanical simulations, the semi-implicit method can *(i)* compute joint angles with the fast explicit scheme and *(ii)* calculate angular velocities with the implicit scheme to stabilize abrupt amplitude changes. Although beneficial for model execution speed, the semi-implicit algorithm may be sensitive to the numerical noise [58]. RK is a typical explicit method and another commonly used integrator. The specific feature of this method is that it solves ODE using a weighted average between four slopes—one at the beginning, two in the middle, and one at the end of the time interval $\Delta t$,—improving the overall slope estimate [59, p. 16]. Although decreasing the integration error, RK requires additional computations per iteration, slowing down the simulation. In addition, as all explicit integration schemes, this method performs poorly for stiff ODEs solved with large integration time steps. Our results support the use of

the implicit Euler over the explicit Euler and RK methods (Figs 3 and 4). A modern state-of-the-art physical engine MuJoCo uses RK, implicit Euler, and semi-implicit Euler modified to be fully implicit in the velocity integration [26]. While this method was not directly analyzed, our results provide insight into its performance advantages.

As we demonstrate in Figs 3 and 6, the solver selection is essential for optimizing a solution to a specific forward dynamic problem. To model human movement, we recommend considering implicit formulations or enforcing the explicit methods with viscoelastic impedance to balance simulation speed, stability, and accuracy (Figs 5, 6 and S2 Fig).

## Simulation frequency

Choosing an optimal simulation frequency is a speed-accuracy tradeoff problem. We solved it by computing inverse-forward dynamics for a range of simulation frequencies and assessing accuracy and execution times. The inverse and forward simulation were performed sequentially, and the numerical errors from both simulations accumulated, as in a similar study [60]. The simulation frequency needs to be minimized with an acceptable level of errors for closed-loop control applications. We chose the error thresholds (<1% kinematic and <5% kinetic) based on our previous evaluation of errors in the approximation of musculoskeletal dynamics [21, 22]. We found forward and inverse computations to be real-time accurate at frequencies as low as 200 Hz (Figs 3 and 5, also see S4 Fig). In simulations of object manipulation, the physics engine *MuJoCo* could maintain stability even at lower simulation frequencies (62.5 Hz) due to many elegant methodological choices [29]. This suggests further benefits of using hybrid solvers; however, the confounding implementation overhead may be difficult to evaluate in software packages.

## Viscoelastic impedance

Passive mechanical impedance can stabilize physical simulation. Modeled using joint spring-dampers, they generate viscoelastic forces to oppose abrupt changes in position and velocity [61]. However, in inverse simulations, additional impedance may lead to an overestimation of joint forces. We used this tradeoff to constraint our optimization and find the joint impedance that both reduced kinematic errors and minimized force distortion. Our analyses showed that mechanical stabilization did not improve the accuracy of implicitly computed kinematics, but, in general, improved the results of the explicit methods. As a result, the hypothesized increase in simulation stability and the reduction in transient kinematic perturbations were true for the explicit but not for the implicit integrators. It is important to note, however, that the impedance-mediated stabilization is similar to kinematic filtering with the correction to the middle of range of motion set by spring rest length. Here, we applied the standard recommendation for filtering kinematics [37] and observed benefits of this approach. It is likely that in simulations that include abrupt mechanical interactions the impedance-mediated stabilization may augment both implicit and explicit methods.

The concept of impedance-mediated stabilization in our study is applied to reducing numerical errors, but it is also theoretically similar to the physiological mechanism. Muscle intrinsic properties and co-contraction generate joint impedance to resist external perturbations. Moreover, the impedance was hypothesized to be the controlled variable for movement and posture control [62–64]. In our previous work, adding expected joint impedance to the musculoskeletal dynamic transformations from joint angles to computed muscle excitations generated activation profiles matching experimentally recorded electromyography [65, p. 63]. While the impedance control has been a controversial theory of motor control in its pure form, the mechanical impedance is an essential feature of muscle coordination. Future studies

could extend our implementation to improve simulations of body movement with realistic muscle-driven impedance. Furthermore, this approach may provide an opportunity for improving the state-of-the-art techniques for controlling active myoelectric prosthetics and orthotics used for rehabilitation of movement [66].

## Limitations

This study was limited to the analysis of the swing phase. The full step analysis including stance is computationally costly typically resolving equations for viscoelastic ground contact with impulse-based velocity-stepping methods, briefly summarized in [29]. Instead, we simulated the standing leg kinematically by attaching it to the ground with a 3-DOF primitive driven by foot segment angles. This simplification introduced kinematic inaccuracies due to the foot roll over the point of contact. The model can be improved in the future by adding translation degrees of freedom and segmenting the foot. Also, the ankle joint was modeled with only rotational DOFs, and the translations could also improve accuracy [67]. Another limitation was the assumption of constant viscoelastic parameters. The time-varying impedance could improve simulation stability beyond the constant values. The magnitude of viscoelastic coefficients could potentially increase to consider muscle co-contraction and further improve stability even at low simulation frequencies. Lastly, we did not simulate forward dynamics with the semi-implicit Euler method because it was not available in Simulink Multibody software. Including this method in the optimal solver search may further improve the speed-accuracy tradeoff for specialized biomechanical models.

## Conclusions

We developed a high-dimensional biomechanical model and an optimization to test viscoelastic stabilization with multiple numerical solver methods and a range of simulation frequencies. We concluded that for the real-time applications, the implicit Euler method with 5 ms time step (200 Hz sampling frequency) and no viscoelastic contribution was optimal. The simulation was kinematically (error <1%) and kinetically (error <5% peak-to-peak) accurate in real time. However, the *viscoelastic impedance was essential* to achieve accurate forward dynamic solutions with explicit solvers.

## Supporting information

**S1 Fig. Mathematical model of the human foot inertia.** (A) Dimensions and centers of mass of the right circular cones; (B) secondary axes used to derive inertia of the frustum; (C) frustum dimensions used to set up a reference frame; (D) frustum modeling human foot with the reference frame originating at the ankle joint.
(PNG)

**S2 Fig.** Kinetic (a) and kinematic (b) errors are affected by stabilizing impedance differently. The errors are shown for the hip adduction-abduction, hip internal external rotation, ankle inversion-eversion, and ankle internal external rotations DOFs of the swinging leg during one representative swing phase simulated with the implicit Euler method at 200 Hz. The kinetic performance decreases at high values, and the kinematic performance decreases at low values. The errors obtained with optimal (*k*,*b*), specific to DOF, sampling frequency, and solver, are squared in light blue. Abbreviations: ad-ab—adduction-abduction; int-ext—internal-external rotation; inv-ever—inversion-eversion.
(PNG)

**S3 Fig.** Stabilizing impedance has dissimilar effects on (a,c) kinetic and (b,d) kinematic errors across DOFs and solver types. The errors are shown for the hip flexion-extension, hip adduction-abduction, hip internal-external rotation, knee flexion-extension, ankle flexion-extension, ankle inversion-eversion, and ankle internal-external rotations DOFs of the swinging leg during one representative swing phase simulated with the built-in recursive (a,c), 4$^{th}$ order Runge-Kutta (b), and explicit Euler (d) methods at 200 Hz. The kinetic performance decreases at high values, and the kinematic performance decreases at low values. The errors obtained with optimal ($k$,$b$), specific to DOF, sampling frequency, and solver, are squared in light blue. Notice a hundredfold difference in scale between values in Fig 2B and S2B Fig. Abbreviations: f-e—flexion-extension; ad-ab—adduction-abduction; int-ext—internal-external rotation; inv-ever—inversion-eversion.
(PNG)

**S4 Fig.** Forward (A) and inverse (B) simulations accuracy are shown as a function of sampling rate, viscoelastic contribution, and numerical solver. The simulations in (A) were solved using numerical integrators: EE—explicit Euler method, RK—4th order Runge-Kutta method, and IE—implicit Euler method. Corresponding inverse simulations (B) were solved with a built-in recursive solver—BR. Labels [k,b] marked the error distributions obtained with optimal impedance.
(PNG)

**S1 File.**
(DOCX)

## Acknowledgments

We thank Trevor Moon for his insightful critique on model implementation and Matthew Yough for proofreading the article.

## Author Contributions

**Conceptualization:** Serhii Bahdasariants, Sergiy Yakovenko.

**Data curation:** Serhii Bahdasariants, Ana Maria Forti Barela, Valeriya Gritsenko, Odair Bacca, José Angelo Barela, Sergiy Yakovenko.

**Formal analysis:** Serhii Bahdasariants.

**Funding acquisition:** Ana Maria Forti Barela, Valeriya Gritsenko, Sergiy Yakovenko.

**Investigation:** Serhii Bahdasariants, Ana Maria Forti Barela, Valeriya Gritsenko, Odair Bacca, José Angelo Barela, Sergiy Yakovenko.

**Methodology:** Serhii Bahdasariants, Sergiy Yakovenko.

**Project administration:** Ana Maria Forti Barela, Sergiy Yakovenko.

**Resources:** Serhii Bahdasariants, Odair Bacca, Sergiy Yakovenko.

**Software:** Serhii Bahdasariants, Valeriya Gritsenko, Sergiy Yakovenko.

**Supervision:** Ana Maria Forti Barela, Sergiy Yakovenko.

**Validation:** Serhii Bahdasariants, Sergiy Yakovenko.

**Visualization:** Serhii Bahdasariants.

**Writing – original draft:** Serhii Bahdasariants, Sergiy Yakovenko.

**Writing – review & editing:** Serhii Bahdasariants, Ana Maria Forti Barela, Valeriya Gritsenko, Odair Bacca, José Angelo Barela, Sergiy Yakovenko.

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
