## [Decision Letter · Decision Letter 0]

27 Mar 2023

PONE-D-23-03468

Does joint impedance improve dynamic leg simulations with explicit and implicit solvers?PLOS ONE

Dear Dr. Yakovenko,

Thank you for submitting your manuscript to PLOS ONE. After careful consideration, we feel that it has merit but does not fully meet PLOS ONE’s publication criteria as it currently stands. Therefore, we invite you to submit a revised version of the manuscript that addresses the points raised during the review process.

ACADEMIC EDITOR:

Both reviewers have identifed some minor issues and have made suggestions to improve. The paper can be accepted after minor revisions.

We look forward to receiving your revised manuscript.

Kind regards,

Noman Naseer, PhD

Academic Editor

PLOS ONE

Journal Requirements:

"SY #1 R03 HD099426-01A1, NIH; AB, VG, SY #2018/04964-8, FAPESP. The sponsors played no role in the study design, data collection and analysis, decision to publish, or preparation of the manuscript."

"This work was funded by the NIH Award (1 R03 HD099426-01A1) and FAPESP Grant (2018/04964-8). We thank Trevor Moon for his insightful critique on model implementation and Matthew Yough for proofreading the article."

"SY #1 R03 HD099426-01A1, NIH; AB, VG, SY #2018/04964-8, FAPESP. The sponsors played no role in the study design, data collection and analysis, decision to publish, or preparation of the manuscript."

Reviewers' comments:

Reviewer's Responses to Questions

**Comments to the Author**

1. Is the manuscript technically sound, and do the data support the conclusions?

Reviewer #1: Yes

Reviewer #2: Yes

2. Has the statistical analysis been performed appropriately and rigorously? 

Reviewer #1: Yes

Reviewer #2: Yes

3. Have the authors made all data underlying the findings in their manuscript fully available?

Reviewer #1: Yes

Reviewer #2: Yes

4. Is the manuscript presented in an intelligible fashion and written in standard English?

Reviewer #1: Yes

Reviewer #2: Yes

5. Review Comments to the Author

Reviewer #1: The present study is well-written and provides useful information for researchers who are interested in dynamic simulation of human movement. My main concern lies in Section D (lines 185-203). I am a bit confused about the "optimization" problem stated in this section. Did the authors actually solve any optimization problems to find the optimal k and b? Or they just found the minimal integrated kinematic and kinetic error (line 198) from 25 simulations across the ranges of k and b specified in lines 191-192? If the latter is the case, I won't recommend calling it an optimization problem since no actual searching is performed. It seems to me the authors have solved a sensitivity problem not an optimization problem.

My specific comments are:

1. Lines 84-85: Shouldn't it be "fail with the increase in the simulation time step"?

2. Eqs (1) and (2): What does each superscript represent in these two equations? For example, what does [NxKxB] mean in Eq. (1)? Please clarify them in the revision.

3. Results section: To help readers better grasp the effects of various choices on the accuracy and stability of the simulation, I would suggest the authors providing a table in the revision to summarize the main findings.

4. Lines 219-220: Have the authors tried to simulate leg swing with the factors that are not optimal? For instance, I will be curious to know the effect of the worst selection of the factors on the accuracy of kinematics and kinetics. Will the resulting kinematics and kinetics errors be much greater than 1% and 5%, respectively? Such information might be useful to researchers having difficulties accessing the optimal choices.

5. Fig.2b: Is angle RMSE actually reported in degree? A RMSE of 5x10^-2 degrees seems very small to me. Just want to confirm it.

6. Fig. 6: Please indicate the sign of each angle in the figure.

7. Fig. S2b: Only two legends are provided here but more than two colors are displayed. Please provide the legends for the other colors.

Reviewer #2: The paper proposed a biomechanical framework to test viscoelastic joint constraints and numerical solver types to decrease simulation errors at low sampling rates. Viscoelastic joint impedance to smooth the transient kinematic perturbations due to the large integration time step was used to increase simulation stability. The method was tested on a human bipedal model. The manuscript is well written and structured.

However, following suggestion may improve the quality of paper; Discussion section need to be improved. The authors have well described and explained the results but additionally proposed work should be compared with benchmark studies. Moreover English of the manuscript need to be improved as academic writing, and formatting of the paper need to be reviewed as Figure numbering should be corrected, Fig 2A and 2B comes after Fig 3-4 and 5 in text citations.

6. PLOS authors have the option to publish the peer review history of their article (what does this mean?). If published, this will include your full peer review and any attached files.

Reviewer #1: **Yes: **Yi-Chung Lin

Reviewer #2: **Yes: **Syed Hammad Nazeer Gilani

---

## [Author Response · Author response to Decision Letter 0]

11 May 2023

Thank you for your comments. We have submitted the point-by-point responses and improved the original submission.

---

## [Decision Letter · Decision Letter 1]

15 Jun 2023

Does joint impedance improve dynamic leg simulations with explicit and implicit solvers?

PONE-D-23-03468R1

Dear Dr. Yakovenko,

We’re pleased to inform you that your manuscript has been judged scientifically suitable for publication and will be formally accepted for publication once it meets all outstanding technical requirements.

Kind regards,

Noman Naseer, PhD

Academic Editor

PLOS ONE

Additional Editor Comments (optional):

Reviewers' comments:

Reviewer's Responses to Questions

**Comments to the Author**

1. If the authors have adequately addressed your comments raised in a previous round of review and you feel that this manuscript is now acceptable for publication, you may indicate that here to bypass the “Comments to the Author” section, enter your conflict of interest statement in the “Confidential to Editor” section, and submit your "Accept" recommendation.

Reviewer #1: All comments have been addressed

Reviewer #2: All comments have been addressed

2. Is the manuscript technically sound, and do the data support the conclusions?

Reviewer #1: Yes

Reviewer #2: Yes

3. Has the statistical analysis been performed appropriately and rigorously? 

Reviewer #1: Yes

Reviewer #2: Yes

4. Have the authors made all data underlying the findings in their manuscript fully available?

Reviewer #1: Yes

Reviewer #2: Yes

5. Is the manuscript presented in an intelligible fashion and written in standard English?

Reviewer #1: Yes

Reviewer #2: Yes

6. Review Comments to the Author

Reviewer #1: The authors have adequately addressed all my comments. Nonetheless, there remain a few minor issues that need their attention. For instance, in Figs 5 and 6, instead of using “n.u.”, it should be mentioned in the figure captions.

Reviewer #2: The Authors have addressed all of my concerns with the original manuscript. The revised manuscript is ready for publication.

7. PLOS authors have the option to publish the peer review history of their article (what does this mean?). If published, this will include your full peer review and any attached files.

Reviewer #1: **Yes: **Yi-Chung Lin

Reviewer #2: **Yes: **Hammad Nazeer

---

## [Editor Report · Acceptance letter]

23 Jun 2023

PONE-D-23-03468R1 

Does joint impedance improve dynamic leg simulations with explicit and implicit solvers? 

Dear Dr. Yakovenko:

I'm pleased to inform you that your manuscript has been deemed suitable for publication in PLOS ONE. Congratulations! Your manuscript is now with our production department. 

Kind regards, 

on behalf of

Dr. Noman Naseer 

Academic Editor

PLOS ONE